# *HER2* Amplification Level Predicts Pathological Complete Response in the Neoadjuvant Setting of HER2-Overexpressing Breast Cancer: A Meta-Analysis and Systematic Review

**DOI:** 10.3390/ijms24043590

**Published:** 2023-02-10

**Authors:** Burak Gonullu, Eurydice Angeli, Frédéric Pamoukdjian, Guilhem Bousquet

**Affiliations:** 1Université Paris Cité, INSERM, UMR_S942 MASCOT, 75006 Paris, France; 2Faculty of Medicine, Yeditepe University, Istanbul 34755, Turkey; 3Université Sorbonne Paris Nord, 9 Avenue Jean Baptiste Clément, 93439 Villetaneuse, France; 4Service d’Oncologie Médicale, Hôpital Avicenne, Assistance Publique Hôpitaux de Paris, 93000 Bobigny, France; 5Service de Médecine Gériatrique, Hôpital Avicenne, Assistance Publique Hôpitaux de Paris, 93000 Bobigny, France

**Keywords:** *HER2* amplification, breast neoplasms, pCR, neoadjuvant therapy, anti-HER2 therapy

## Abstract

Anti-HER2 therapies have dramatically improved the prognosis of human epidermal growth factor receptor 2 (HER2)-overexpressing cancers. However, the correlation between the *HER2* copy number and the response rate to anti-HER2 remains unclear. Here, following the PRISMA method, we performed a meta-analysis in the neoadjuvant setting in breast cancer to study the association between the *HER2* amplification level and the pathological complete response (pCR) to anti-HER2 therapies. Nine articles (four clinical trials, five observational studies) were retrieved after full-text screening, involving 11,238 women with locally advanced breast cancer in the neoadjuvant setting. The median *HER2/CEP17* ratio cut-off value was 5.0 ± 5.0 (min-max = 1.0–14.0). For the overall population, the median pCR rate was 48% using the random effect model. The studies were categorized in quartiles as follows: ≤2 (Class 1); 2.1 to 5.0 (Class 2); 5.1 to 7.0 (Class 3); and >7.0 (Class 4). After grouping, the pCR rates were 33%, 49%, 57%, and 79%, respectively. When we excluded the study by Greenwell et al., which accounted for 90% of the patients, using the same quartiles, we still observed an increasing rate of pCR as the *HER2/CEP17* ratio increased. This is the first meta-analysis demonstrating the relationship between the *HER2* amplification level and the percentage of pCR in the neoadjuvant setting among women with HER2-overexpressing breast cancer, with potential therapeutic applications.

## 1. Introduction

Human epidermal growth factor receptor 2 (HER2)-overexpressing cancers, mainly breast and gastric cancers, are significant public health concerns. For breast cancer, when the *HER2* gene is amplified (either a high copy number or polysomy 17), it most constantly leads to an increased mRNA expression and then to HER2 protein overexpression on the surface of cancer cells [1], which promotes their growth and division [2]. HER2 overexpression is found in approximately 15% of breast cancers [3] and was initially associated with a poorer prognosis and increased risk of recurrence [4]. For 20 years, anti-HER2 therapies have dramatically improved the prognosis of HER2-overexpressing cancers, particularly breast and gastric cancers [5]. Anti-HER2 therapies comprise tyrosine kinase inhibitors and monoclonal antibodies, such as trastuzumab, which binds to the HER2 membrane receptor and inhibits its activity. Trastuzumab, the leading anti-HER2 antibody, was first approved for metastatic HER2-overexpressing breast cancer [6], and then in the adjuvant setting [7]. The combination of trastuzumab and cytotoxic chemotherapy, mainly taxanes, has been shown to significantly improve outcomes for women with HER2-overexpressing breast cancer, improving time to disease progression and overall survival [6]. Since then, several anti-HER2 have been approved, such as pertuzumab, another monoclonal antibody that inhibits HER2 receptor heterodimerization with another HER receptor family. In the metastatic setting, the addition of pertuzumab to a combination of trastuzumab and docetaxel has considerably improved patient prognosis with an 8-year survival of 37% [8]. Additionally, tyrosine kinase inhibitors such as lapatinib and tucatinib have been developed, as well as more recent innovations; for example, antibody–drug conjugates such as T-DM1 or trastuzumab deruxtecan [9,10,11,12,13,14,15,16,17,18].

In the neoadjuvant setting, several clinical trials have been conducted to evaluate the effectiveness of anti-HER2 therapies for localized breast cancer, using the same anti-HER2 therapies initially approved in the metastatic setting [19,20,21,22,23]. Neoadjuvant chemotherapy has become standard care for locally advanced breast cancer, enabling more frequent breast preservation surgeries. In addition, neoadjuvant setting is an ideal situation to assess response to treatment, since the pathological complete response (pCR) after neoadjuvant chemotherapy is a validated surrogate marker predicting overall survival in the treatment of locally advanced breast cancer [24]. Different protocols have been implemented combining anti-HER2 with standard chemotherapies with or without anthracyclines due to cumulative cardiac toxicity when combined with anti-HER2. Dual anti-HER2 therapies have enabled a significant increase in the percentage of pCRs up to 70%, leading to the approval of the combination of trastuzumab, pertuzumab, and taxane as standard care in this setting [23,25,26], even if an overall survival benefit has not yet been demonstrated.

There remains an ongoing debate and uncertainty regarding the appropriate testing to determine *HER2* status and inform treatment decisions. Some studies have shown a quasi-complete concordance between immunohistochemistry (IHC) and in situ hybridization (ISH) in more than 97% of cases [1,27]. In contrast, in a meta-analysis on 6629 women with breast cancers, 9% with a 3+ IHC score (HER2 overexpression) were not amplified using ISH, and 4% with a 0/1+ IHC score (HER2 low expression) had a high copy number of the *HER2* gene [28]. These discordances may be linked to technical problems, but also intratumor heterogeneity. Despite this, the decision to apply anti-HER2 therapy is usually guided by HER2 oncoprotein overexpression using IHC, which is thus usually linked to *HER2* amplification, classically defined as ≥4 copies per nucleus and a *HER2/CEP17* ratio of ≥2 using ISH according to the 2018 CAP/ASCO guidelines [29]. Most studies evaluating the effectiveness of anti-HER2 therapies have primarily relied on IHC to determine *HER2* status, rather than ISH. This is because immunohistochemistry is easier to perform in daily practice. However, by way of using IHC alone, there is no fine subcategorization of cancers with a 3+ IHC score, and the correlation between the *HER2* copy number and the response rate to anti-HER2 thus remains unclear, whatever the clinical setting.

Here, we performed the first meta-analysis to study this association between *HER2* amplification level and pCR to anti-HER2 therapies for the treatment of localized breast cancer in the neoadjuvant setting.

## 2. Materials and Methods

### 2.1. Search Strategy and Selection Criteria

The recommendations of the Preferred Reporting Items for Systematic Reviews and Meta-Analyses (PRISMA) method were followed to report this meta-analysis [30]. The study selection was carried out in two electronic databases (MEDLINE via PubMed and ScienceDirect) by searching articles published in English up to 30 December 2021, using the search algorithm (neoplasm[MesH]) AND (HER2) AND (amplification) AND (Trastuzumab OR pertuzumab OR Trastuzumab-DM1 OR T-DM1 OR Lapatinib OR Trastuzumab deruxtecan OR Tucatinib). The PROSPERO study registration number is CRD42022306320, registered in February 2022.

In line with the PICOS recommendations, the inclusion criteria were: (i) Participants: women aged 18 years and over with HER2-overexpressing breast cancer in the neoadjuvant setting; (ii) Intervention: Anti-HER2-based neoadjuvant therapy whatever the anti-HER2 drug; (iii) Comparator: *HER2* gene amplification level obtained using a quantitative method (FISH, ddPCR, NGS); (iv) Outcome: the pCR outcome reported in the study and available statistical data comparing pCR and the *HER2* gene amplification level; (v) Study design: Randomized clinical trials (RCTs), non-RCTs, and observational cohort studies; Only papers in English were included in this meta-analysis.

The exclusion criteria were studies involving non-HER2-overexpressing breast cancer; studies in the adjuvant or metastatic setting; studies without anti-HER2-based neoadjuvant therapy; studies without details on *HER2* gene amplification level; studies without available statistical data comparing pCR and the *HER2* gene amplification level; studies without pCR outcome reported in the study; and studies other than randomized clinical trials (RCTs), non-RCTs, and observational cohort studies.

Titles were identified by the above-mentioned search algorithm and screened by two authors (BG and EA). Articles were first selected based on titles and abstracts, then evaluated by a perusal of the full text according to the inclusion criteria. All excluded studies were recorded and reasons for exclusion were accounted for. Any debate concerning inclusion during the full-text screening was resolved by discussion and consensus. If several publications on the same trial were retrieved, only the publication with the largest number of patients (and the most informative) was included.

Any disagreements were resolved by discussion. Data retrieved from the publications included the following: year of publication, first author’s name, country, sample size, disease stage, *HER2/CEP17* ratio, and the rate of pCR. The *HER2/CEP17* ratio was determined using FISH. If only the *HER2* gene copy number was assessed, it was divided by 2 to obtain the *HER2/CEP17* ratio. Patients with ≥4 copies per nucleus and a *HER2/CEP17* ratio of <2 corresponded to patients with HER2 overexpression linked to polysomy 17. One study in the meta-analysis included such patients [31] who were classified in the *HER2/CEP17* category of ≤2. Patients with <4 copies per nucleus but a *HER2/CEP17* ratio of ≥2 corresponded to patients with monosomy 17, but no data related to such patients was available in the studies included in our meta-analysis.

Pathological complete response was defined as a complete pathological remission of invasive tumor cells in both the breast and the axillary lymph nodes [32].

The quality of the eligible studies was assessed using the Cochrane Handbook for Systematic Reviews of Interventions. Version 2 of the Cochrane risk-of-bias tool for randomized trials (RoB 2) and The Risk Of Bias In Non-randomized Studies of Interventions (ROBINS-I) were used to assess the risk of bias. In all cases, two authors independently assessed the risk of bias among the studies included, and any disagreements were resolved by discussion to reach a consensus. We incorporated the results of the risk of bias assessment into the review using standard tables, systematic narrative descriptions, and commentaries about each of the elements, leading to an overall assessment of the risk of bias in the studies included and an assessment of the internal validity of the review results.

### 2.2. Statistical Analysis

The data were analyzed using R statistical software (version 4.1.0; R Foundation for Statistical Computing, Vienna, Austria; http://www.r-project.org, accessed on 20 February 2022). Categorical variables were summarized as numbers (percentages), and continuous variables were summarized as medians ± interquartile range (IQR). On the basis of the selected articles, we performed a meta-analysis (with the package “meta”) to assess the relationship between the *HER2/CEP17* ratio and pathological complete response (%). The median cut-off value for the *HER2/CEP17* ratio was 5.0 ± 5.0 (min-max = 1.0–14.0). Based on sensitivity analyses [31,33], we categorized the studies in quartiles, as follows: ≤2 (Class 1); 2.1 to 5.0 (Class 2); 5.1 to 7.0 (Class 3); and >7.0 (Class 4). We also implemented categorization in tertiles, as follows: ≤2 (Class 1); 2.1 to 6.0 (Class 2); and >6.0 (Class 3). We assessed the heterogeneity of study results using the I2 indicator and the Cochran’s Q test. I2 values of 0%, 25%, 50%, and 75% were considered to indicate none, low, moderate, and high heterogeneity, respectively. A *p* value of ≤0.05 on the Q-test indicated significant heterogeneity. Because of a significant heterogeneity across studies, the pooled results were summarized graphically as proportions with 95% confidence intervals (95% CI) in a forest plot using a random effect model. Publication bias was assessed graphically using a funnel plot and quantitatively using Egger’s test. All tests were two-sided, and the threshold for statistical significance was set at a *p*-value of under 0.05.

## 3. Results and Discussion

The literature search and screening processes are detailed by a PRISMA flow diagram in Figure 1.

We identified 905 articles, and 13 additional articles from the citation search. After excluding duplicate articles and screening by title and abstract, we were left with 112 articles for full-text assessment. We then excluded a further 103 articles, most of which did not assess the outcomes of interest (49), or did not report *HER2* amplification levels (35). Nine articles were finally retrieved. These nine studies (four clinical trials, five observational studies) included 11,238 women with locally advanced breast cancer in the neoadjuvant setting, with the study by Greenwell et al. accounting for 90% of these patients [31] (Table 1). Mean/median ages ranged from 46 to 51 years. Overall, various anti-HER2 regimens were used as neoadjuvant chemotherapy. The anti-HER2 treatment used was trastuzumab monotherapy in six studies [33,34,35,36,37,38], and the combined treatment of trastuzumab and lapatinib in two studies [39,40]. For the study by Greenwell et al., there were no details about the type of anti-HER2 therapy. We assumed that the women only received trastuzumab-based chemotherapy since they were treated before 2013 [41]. All studies used fluorescent in situ hybridization (FISH) to assess the *HER2* amplification level. The pCR definition was clear for all studies but one. Four studies used the definition of ypT0/is ypN0, three studies used the definition of ypT0 ypN0, and one study used the definition of ypT0-is.

Regarding other clinico-biological characteristics, the estrogen receptor status was available for four studies, with a positivity ranging from 20% to 76%. There was limited data on cancer type, which was only reported in two studies, with most cancers being described as ductal type. For TNM staging, the tumor size was reported in five studies, with a proportion T3-T4 stages ranging from 23% to 67%. Only three studies reported the percentage of women without lymph node involvement (N0), up to 55% in the study by Singer et al. Two additional studies reported the percentage of stage III tumors in up to 38% of the patients.

There was a significant publication bias among the nine studies (*p* = 0.02) (Figure 2), except after the exclusion of the study by Greenwell et al. (*p* = 0.06) (Appendix A). The overall quality of the studies was good (Appendix A).

The median cut-off value for the *HER2/CEP17* ratio was 5.0 ± 5.0 (min-max = 1.0 −14.0). In our analysis, we divided the patient groups that we collected into distinct subgroups based on the cutoff values obtained from the included studies. We then incorporated the pCR data from these studies into the subgroups to create a forest plot, which enabled us to visualize the overall distribution of pCR rates across the different studies. For the overall population, the median pCR rate was 48% using a random effect model. However, the analysis also revealed significant heterogeneity across the studies.(I^2^ = 95%, *p* < 0.01) (Figure 3).

We then implemented a categorization in tertiles of the *HER2/CEP17* ratio as follows: ≤2 (Class 1); 2.1 to 6.0 (Class 2); and >6.0 (Class 3). The pCR rates were 33%, 53%, and 67%, respectively (Appendix A). We further categorized studies in quartiles: ≤2 (Class 1); 2.1 to 5.0 (Class 2); 5.1 to 7.0 (Class 3); and >7.0 (Class 4). The pCR rate was even greater for a *HER2/CEP17* ratio of >7.0 (79%) and the heterogeneity was lower (I^2^ = 0%, *p* = 0.75) (Figure 4).

The Class 1 ratio was ≤2 because 4848 women included in the study by Greenwell et al. had HER2-overexpressing breast cancer with a *HER2/CEP17* ratio of ≤2 and a *HER2* copy number of >6, corresponding to polysomy 17. These 4848 women had a pCR of 29.2%. When we excluded these 4848 women, the new categorization in quartiles (Class 1: ≥2, Class 2: ≥3, Class 3: ≥6, and Class 4: ≥9) did not change the mean/median pCR in the different classes (Appendix A).

When we excluded the study by Greenwell et al., using the same quartiles, we still observed an increasing rate of pCR as the *HER2/CEP17* ratio increased (Class 1 = 34%, Class 2 = 51%, Class 3 = 65%, Class 4 = 92%). The exclusion of this study also reduced the overall heterogeneity (I^2^ = 71%) (Figure 5).

This is the first meta-analysis demonstrating the relationship between an increased *HER2/CEP17* ratio and the percentage of pCR in the neoadjuvant setting across a population of 11,238 women with HER2-overexpressing breast cancer. For twenty years, in the neoadjuvant setting, there have been numerous clinical trials assessing the benefit of using anti-HER2 therapies for the treatment of locally advanced HER2-overexpressing breast cancers. In the first reported clinical trial, in 2005, the patients receiving a combination of trastuzumab plus standard chemotherapy had a 65% pCR [42]. Since then, several clinical trials have attempted to optimize chemotherapy regimens, in particular using dual blockade with the combination of trastuzumab and pertuzumab [23]. In the NeoSphere trial published in 2012, the combination of trastuzumab, pertuzumab, and docetaxel significantly improved pCR compared to a combination of trastuzumab and docetaxel (from 29% to 45.8%) [23]. In our meta-analysis, we found that the higher the level of *HER2* amplification, the higher the pCR after trastuzumab-based neoadjuvant chemotherapy, reaching 79% when the *HER2/CEP17* ratio was above 7. A previous meta-analysis from three studies and 1360 patients in the adjuvant setting failed to show any significant relationship between the *HER2* amplification level and disease-free survival [43]. In contrast, in metastatic settings when the tumor burden is much larger, some studies have evidenced a relationship between *HER2* amplification level and response to treatment or survival parameters [44,45]. We chose the neoadjuvant setting as it is an ideal situation to accurately assess the response to therapy using a consensual biomarker, pCR. Most studies usually consider ypT0/is ypN0, which is the absence of invasive cancer cells in the primary tumor and in the axillary lymph nodes [24]. The association between pCR and improved survival parameters has been largely demonstrated. In a meta-analysis across a population of 5768 patients with HER2-overexpressing breast cancer in the neoadjuvant setting, pCR was significantly associated with improved event-free survival and overall survival, with respective hazard ratios of 0.37 and 0.34 [46]. In our meta-analysis, we could not assess the relationship between *HER2* amplification level and survival parameters, since the available data was limited to two studies and 561 patients [36,37].

The stringent methodology was a strength of our meta-analysis, with clearly defined inclusion criteria and a rigorous selection of the studies. Despite this, the number of nine studies retrieved was a limitation, and we had to exclude most of the 112 initially eligible studies because the quantitative data on pCR or *HER2* amplification level was lacking. In addition, most clinical trials using anti-HER2 treatments were not included in this meta-analysis. When we searched PubMed with the keywords “((trastuzumab) AND (breast cancer)) AND (neoadjuvant)” and the filter “clinical trial”, we obtained 282 results, and most studies only used immunohistochemistry to assess *HER2* status without any data on the *HER2* copy number. A systematic assessment of the *HER2/CEP17* ratio and *HER2* copy number would undoubtedly add a new perspective to address the proportional link between *HER2* amplification level and the response to anti-HER2 treatment. In our meta-analysis, we did not have the *HER2/CEP17* ratio and *HER2* copy number for all nine studies. Two of them only reported gene copy number while two others only reported *HER2/CEP17* ratio. We chose the *HER2/CEP17* ratio, dividing the gene copy number by two, assuming that this is more accurate than extrapolating the copy number from the *HER2/CEP17* ratio. Indeed, there might be some patients with < 4 copies per nucleus but a *HER2/CEP17* ratio of ≥2, corresponding to patients with monosomy 17. In contrast, dividing the gene copy number by two, we might have re-classified some patients since patients with HER2 overexpression linked to polysomy 17 have ≥4 copies per nucleus but a *HER2/CEP17* ratio of <2 [29]. We did this for the two studies by Guiu et al. and Arnould et al. These articles included a total of 170 patients which only represents 1.5% of the total patient population (*n* = 11,238) included in the meta-analysis.

Another limitation of our meta-analysis was linked to the fact that the study by Greenwell et al. accounted for 90% of the patients. However, even after the exclusion of this study, the results remained positive.

In our meta-analysis, data was also limited to identify a link between the *HER2* gene copy number and clinical or biological parameters. In their meta-analysis across a population of 5768 patients with HER2-overexpressing breast cancers, Broglio et al. reported that the association between pCR and improved survival was greater in the case of hormone-receptor negative status compared to hormone-positive tumors (HR = 0.29 vs. 0.52) [46]. In the WSG-ADAPT phase II trial, patients with HER2-overexpressing breast cancer and hormone-negative receptor treated with trastuzumab, pertuzumab, and paclitaxel achieved a very high pCR rate of 78.6% (ypT0 ypN0) [47]. We could not test the same association due to limited available data. However, in their study, Greenwell et al. also reached similar conclusions, showing a gradually increased effect with the *HER2* copy number, both in the sub-group of women with hormone-receptor positive and negative cancers [31]. Even in the case of a high *HER2/CEP17* ratio of >7, the pCR rate is 9–15% higher for hormone-receptor negative cancers [31].

Our results have clinical value since pCR has proved to be an appropriate surrogate marker predicting survival for women treated in the neoadjuvant setting [24]. In our meta-analysis, almost all women received trastuzumab-based single blockade. Indeed, out of a total patient population of 11,238, only 99 individuals (0.88% of the total) received dual therapy as part of their treatment. Despite the fact that most women were treated with a single trastuzumab blockade, pCR rates ranged from 33% to 79% depending on the *HER2* amplification level. For 10 years, dual HER2 blockade has been widely used in the neoadjuvant setting, improving pCR and relapse-free survival compared to trastuzumab-based single blockade chemotherapy. However, the overall survival benefit of dual HER2 blockade remains debated [48,49], possibly because the patient selection was made blind to the *HER2* amplification level. In a meta-analysis involving 15,284 patients, pCR rates for the dual-therapy group and the mono-therapy group were 51.60% and 38.26%, respectively [50]. In another meta-analysis on 1410 patients, dual HER2 blockade with trastuzumab and lapatinib led to a significantly improved overall survival, with pCR rates up to 63% [51]. In our study, in the sub-group of 1089 patients with a *HER2/CEP17* ratio above 7, up to 13 patients in the study by Hurvitz et al. were treated with a combination of trastuzumab and lapatinib, with an 85% pCR rate. In the sub-group of women with a *HER2/CEP17* ratio above 7, the pCR rate reached 79%, questioning the benefit of dual blockade in this subgroup to avoid unnecessary toxicities. Indeed, dual anti-HER2 blockade, despite usually being well tolerated, significantly increases toxicities, in particular with severe diarrhea leading to frequent treatment discontinuation [52,53]. Several de-escalating strategies have been tested in the neoadjuvant setting to decrease toxicities, mainly omitting the use of chemotherapy agents, with disappointing results. In the 107 patients of the NeoSphere trial treated with the combination of trastuzumab and pertuzumab without chemotherapy, the pCR was only 16.8% [23]. In the WSG-ADAPT phase II trial, 5-year invasive disease-free survival decreased from 98% to 87% when omitting paclitaxel from the combination with trastuzumab and pertuzumab [47]. According to the results of our meta-analysis, the de-escalating strategy could take into account the *HER2* copy number in order to discuss a single anti-HER2 therapy combined with a cytotoxic drug instead of a dual anti-HER2 therapy, particularly in the case of a high *HER2* amplification level.

In contrast, the pCR rate was low in our study when the *HER2/CEP17* ratio was between 2 and 3, at 34%. Dual HER2 blockade, but also antibody–drug conjugates such as trastuzumab-deruxtecan, could be much more beneficial for these patients with lower *HER2* amplification levels. Trastuzumab-deruxtecan is an antibody–drug conjugate made of a humanized anti-HER2 antibody linked with a topoisomerase I inhibitor of high potency and a high drug-to-antibody ratio of 8 [54]. Trastuzumab-deruxtecan was the first developed in heavily pretreated women with HER2-overexpressing breast cancer with very high response rates [16], and has shown its superiority to T-DM1 in second-line settings with a 12-month progression-free survival of 75% versus 34% [17]. Recently, trastuzumab-deruxtecan has been proven to be efficacious in the metastatic setting even in patients with a low HER2 expression level (score 1+ or 2+ but no FISH amplification) with a 6.4-month absolute overall survival gain (from 17.5 months for women not receiving trastuzumab-deruxtecan to 23.9 months for those who received trastuzumab-deruxtecan) [18]. Trastuzumab-deruxtecan is currently being evaluated in the neoadjuvant setting to assess whether it could replace standard chemotherapy or anti-HER2 therapies.

Further studies are thus required to confirm whether *HER2* amplification level can contribute to patient selection for HER2-targeted therapy by strengthening the decision to use dual HER2 blockade in the neoadjuvant setting.

## 4. Conclusions

Our study demonstrated the correlation between the level of *HER2* amplification and pCR in the neoadjuvant setting with potential therapeutic applications.

Our meta-analysis was the first to assess the correlation between *HER2* amplification levels and pathological complete response for the treatment of HER2-overexpressing early breast cancer. For the past decade, dual HER2 blockade combining trastuzumab, pertuzumab and cytotoxic drugs has been widely used as standard care to increase pCR and thus survival parameters. However, this dose-intense approach might not be beneficial for all patients, as it comes with unnecessary toxicities, and it remains unclear which patient could benefit from de-escalating therapeutic approaches. Our findings provide evidence for the use of *HER2* amplification levels as a predictive marker for treatment response, thus guiding treatment decisions in women with HER2-overexpressing early breast cancer. More specifically, in the case of a high *HER2* amplification level, a single anti-HER2 therapy combined with a cytotoxic drug may be as efficacious as a dual anti-HER2 therapy, but with fewer toxicities. In contrast, in the case of a low *HER2* amplification level, dose-intense approaches should probably be preferred using dual anti-HER2 therapies or promising drugs such as trastuzumab-deruxtecan.

The results of our meta-analysis provide some evidence for the use of *HER2* amplification level, ideally combining *HER2* copy number and *HER2/CEP17* ratio, as an accurate biomarker predicting the percentage of pathological complete response in the neoadjuvant setting. Further studies are required to confirm these findings, particularly to demonstrate whether this biomarker could be used for therapeutic decisions, either dose intensification or de-escalating approaches.

## Figures and Tables

**Figure 1 ijms-24-03590-f001:**
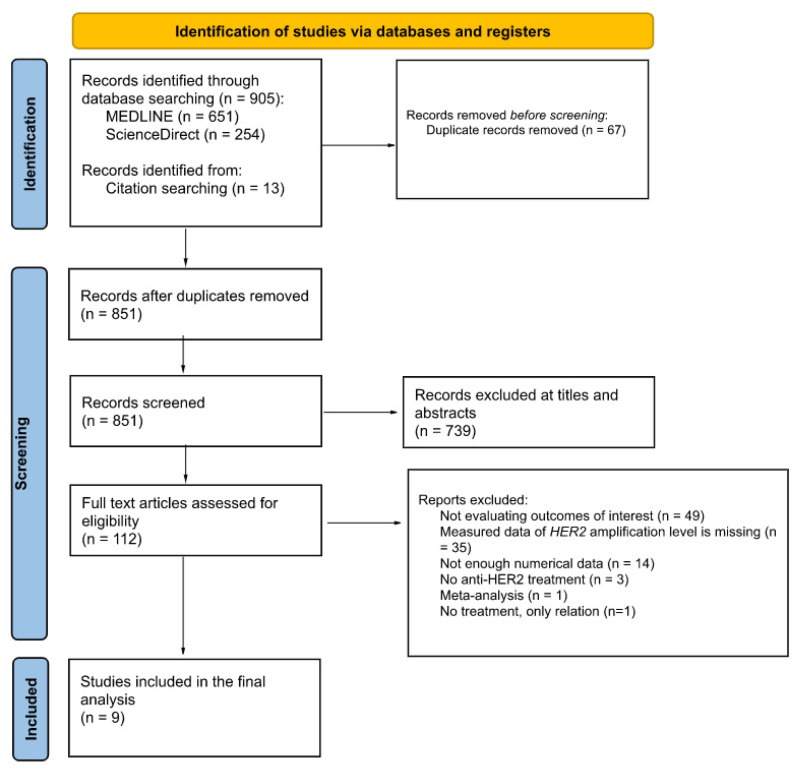
PRISMA Flow Diagram.

**Figure 2 ijms-24-03590-f002:**
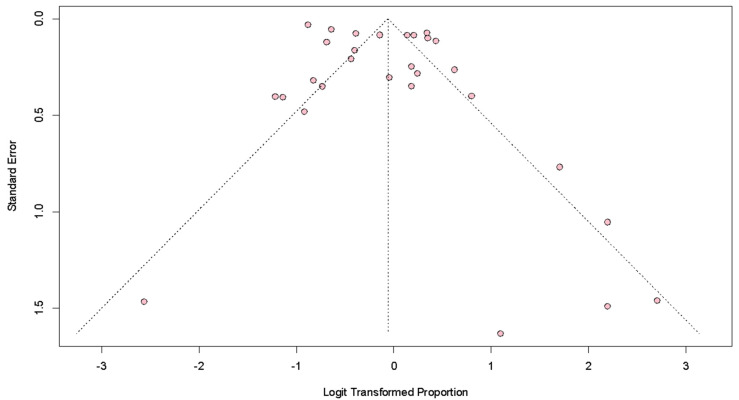
Funnel plot for pathological complete response across the nine studies. *p*-value for publication bias = 0.02.

**Figure 3 ijms-24-03590-f003:**
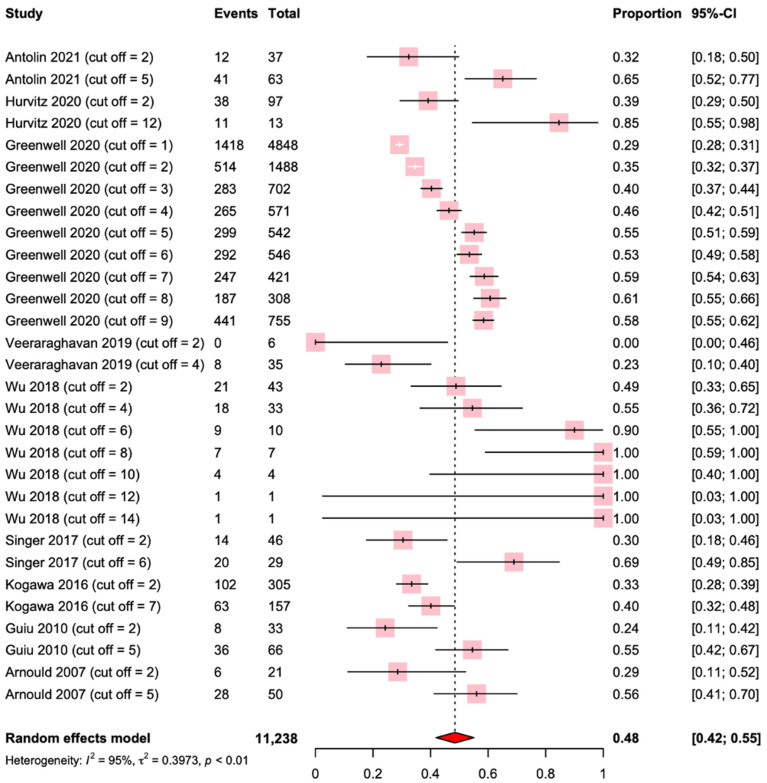
Forest plot for pathological complete response among the nine studies. “Event” refers to the number of patients with a pCR for a determined *HER2/CEP17* ratio. “Total” refers to the total number of patients for the same *HER2/CEP17* ratio. In this case, for the same study, several *HER2/CEP17* ratio thresholds were considered [31,33,34,35,36,37,38,39,40].

**Figure 4 ijms-24-03590-f004:**
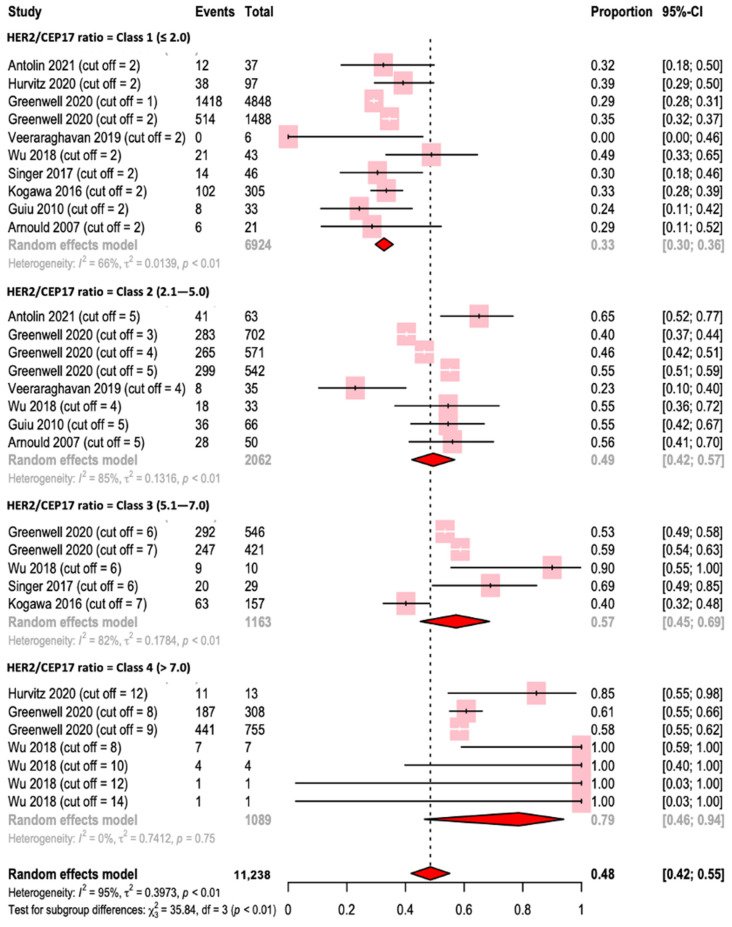
Forest plot for pathological complete response according to the *HER2* interval (quartiles) [31,33,34,35,36,37,38,39,40].

**Figure 5 ijms-24-03590-f005:**
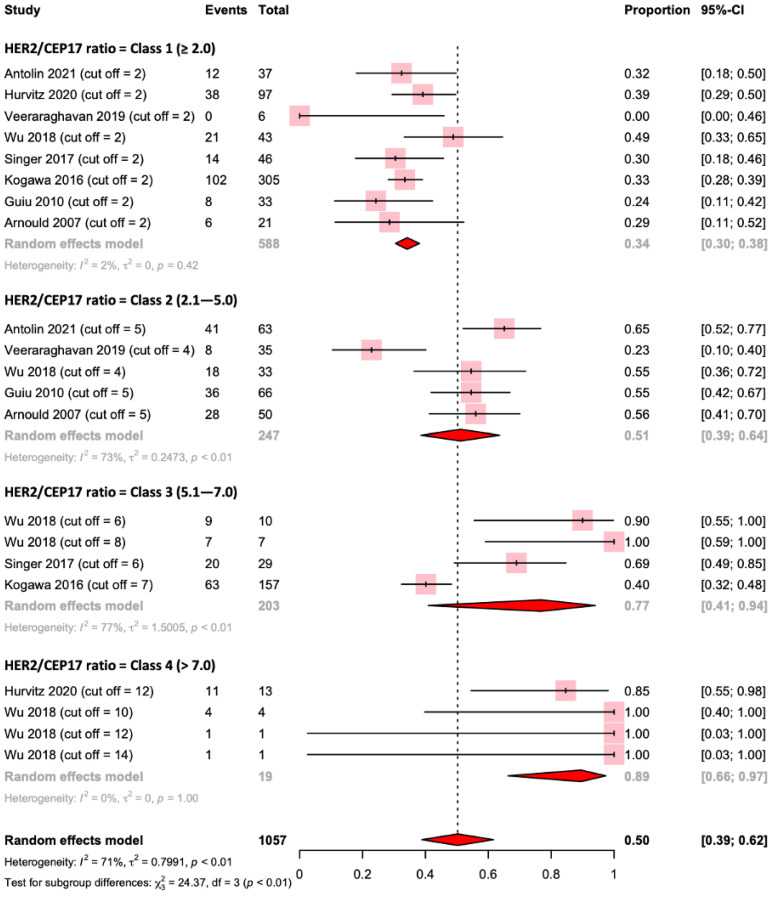
Forest plot for pathological complete response according to the HER2 interval (quartiles) and excluding the study by Greenwell et al. [31,33,34,35,36,37,38,39,40].

**Table 1 ijms-24-03590-t001:** Characteristics of the studies.

Ref.	Date	Study Design	Number of Patients ***	pCR Definition	Median Age	Treatment	Estrogen Receptor (+)	Ductal Type	T3-T4	Stage (III)	N0	Assay Method
Antolín et al. [34]	2021	Observational	100	ypT0/is ypN0	49.5 (30–79)	TaAT	N/A	95%	N/A	N/A	N/A	FISH
Hurvitz et al. [39]	2020	Clinical Trial	110	ypT0/is ypN0	(1) 48 (2) 51 (3) 47	(1) DCT (2) DCL (3) DCTL	(1) 59% (2) 50% (3) 59%	(1) 97% (2) 94% (3) 90%	N/A	(1) 35% (2) 19% (3) 29%	N/A	FISH
Greenwell et al. [31]	2020	Observational	10181	N/A	N/A	Neoadjuvant Chemotherapy	20–55%	N/A	N/A	N/A	N/A	FISH
Veeraraghavan et al. [40]	2019	Clinical Trial	41	ypT0-is	49 (31–74)	TL	N/A	N/A	62%	N/A	N/A	FISH
Wu et al. [33]	2018	Clinical Trial	99	ypT0 ypN0	N/A	PCisT * + Endocrine Tx	76%	N/A	67%	N/A	N/A	FISH
Singer et al. [35]	2017	Clinical Trial	75	ypT0 ypN0	50.3 (25.4–76.9)	(1) E/DT * or E/DCT * (2) DT or DBT or DDoxT or DBDoxT	N/A	N/A	23%	N/A	55%	FISH
Kogawa et al. [36]	2016	Observational	462	ypT0/is ypN0	N/A	ATaT * or AT *	45.7%	N/A	N/A	IIIA = 25% IIIB = 36.9% IIIC = 38%	N/A	FISH
Guiu et al. [37]	2010	Observational	99	ypT0 ypN0	46.6 (26–76) **	DCT or DT	N/A	N/A	23%	N/A	47%	FISH
Arnould et al. [38]	2007	Observational	71	ypT0/is ypN0	46 (27–67) **	DCT or DT	N/A	N/A	28%	N/A	49.5%	FISH

* = if HER2 positive; ** = mean age; *** = number of patients who had HER2 analysis; TaAT = Taxane + Anthracycline + Trastuzumab; DCT = Docetaxel + Carboplatin + Trastuzumab; DCL = Docetaxel + Carboplatin + Lapatinib; DCTL = Docetaxel + Carboplatin + Trastuzumab + Lapatinib; TL = Trastuzumab + Lapatinib; PCisT + Endocrine Tx = Paclitaxel + Cisplatin + Trastuzumab (if HER2 positive) + Endocrine tx (if receptor positive); E/DT = Epirubicine/Docetaxel + Trastuzumab; E/DCT = Epirubicine/Docetaxel + Capecitabine + Trastuzumab (if HER2 positive); DT = Docetaxel + Trastuzumab; DBT = Docetaxel + Bevacizumab + Trastuzumab; DDoxT = Docetaxel + Doxorubicine + Trastuzumab; DBDoxT = Docetaxel + Bevacizumab + Doxorubicine + Trastuzumab; ATaT = Anthracycline + Taxane + Trastuzumab; AT = Anthracycline + Trastuzumab; FISH = Fluorescence in situ hybridization.

## Data Availability

Not applicable (The current study was conducted on the basis of published literature and no datasets were generated.).

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
