# Peer review of "HER2 Amplification Level Predicts Pathological Complete Response in the Neoadjuvant Setting of HER2-Overexpressing Breast Cancer: A Meta-Analysis and Systematic Review"

_ijms, 2023, doi:10.3390/ijms24043590_

Round 1
Reviewer 1 Report
Minor comments:
· I suggest rewriting the last idea in the introduction section: “Neoadjuvant setting being an ideal situation to assess response to treatment, we performed here the first meta-analysis to study the association between HER2 amplification level and pCR to anti-HER2 therapies for the treatment of localized breast cancer.” It is not clear enough!!
· I suggest rewriting this idea in the results section: “The anti-HER2 was trastuzumab in 6 studies [26–31], and the combination of trastuzumab and lapatinib in 2 studies” It is not clear enough!!
· To make easier lecture and comprehension of the results the supplementary figures, in the manuscript, it would be better indicating those figures in order.
· In funnel plot figures I suggest to explain the meaning of “events” and “Total”. The above to clarify the interpretation of the figures and to explain why analyses are repeated within studies that vary in number of events and total.
Major comments:
· Several studies enrolled in this manuscript most probably reported different clinical-pathological characteristics. It would be interesting to meta-analyze some of these features to establish possible association with the categorization proposed for patients with HER2-overexpressing breast cancer (e.g. Stage, Grading, Tumor size). Then, the author can make a conclusion based on this result.
· Authors indicate that included patients with HER2-overexpressing breast cancer. Since they indicate overexpression is defined as ≥ 4 copies per nucleus and a HER2/CEP17 ratio ≥ 2, how were considered patient with ≥ 4 copies per nucleus but a HER2/CEP17 ratio < 2 or patient with < 4 copies per nucleus but a HER2/CEP17 ratio ≥ 2??. I think it is necessary to clarify this point (methods), overall when in the first paragraph of the page 8, it is explained why in most figures there is a class of categorization including cases without HER2-overexpression (ratio <2).
· Regarding the above, do you consider that in these “equivocal cases” the treatment decision could be seen as influenced??. How do you correlate your results with this topic? How do your results may provide insights into the clarification of how these patients should be treated?? Some insight into the above should be included in the discussion section.
· Authors declare that “If only the HER2 gene copy number was assessed, it was divided by 2 to obtain the HER2/CEP17 ratio”. For how many of the included studies do you did it?? What implications for analysis does this have? This must be considered a limitation that deserves to be highlighted in the discussion!!!
“For the study by Greenwell et al., there were no details about the type of anti-HER2 therapy. We assumed that the women only received trastuzumab-based chemotherapy, since they were treated before 2013”. If this were not the case, do you think that the results provided by this study would be different? I suggest discuss this, together with the ideas of single vs dual blockade of HER2; just to better justify your point about toxicity
Author Response
Response to Reviewer 1 Comments
Minor comments:
Point 1: I suggest rewriting the last idea in the introduction section: “Neoadjuvant setting being an ideal situation to assess response to treatment, we performed here the first meta-analysis to study the association between HER2 amplification level and pCR to anti-HER2 therapies for the treatment of localized breast cancer.” It is not clear enough!!
Response 1: To follow the advice by Reviewer 1, we have rewritten the sentence which is now in page 2, (lines 68-70) “We performed here the first meta-analysis to study the association between HER2 amplification level and pCR to anti-HER2 therapies for the treatment of localized breast cancer in the neoadjuvant setting.”
Point 2: I suggest rewriting this idea in the results section: “The anti-HER2 was trastuzumab in 6 studies [26–31], and the combination of trastuzumab and lapatinib in 2 studies” It is not clear enough!!
Response 2: To follow the advice, we have rewritten the sentence which is now in page 5, (lines 164-166) “The anti-HER2 treatment used was trastuzumab monotherapy in 6 studies [26–31], and the combined treatment of trastuzumab and lapatinib in 2 studies [32] [33].”
Point 3: To make easier lecture and comprehension of the results the supplementary figures, in the manuscript, it would be better indicating those figures in order.
Response 3: Supplementary figures are now presented in a clear and organized manner, to facilitate ease of understanding for our readers.
Point 4: In funnel plot figures I suggest to explain the meaning of “events” and “Total”. The above to clarify the interpretation of the figures and to explain why analyses are repeated within studies that vary in number of events and total.
Response 4: "Event" refers to the number of patients with a pCR for a determined HER2/CEP17 ratio. "Total" refers to the total number of patients for the same HER2/CEP17 ratio. In order to analyze the data, we divided the patients into groups based on the predetermined cutoff values as specified in the relevant studies.
To follow the advice by Reviewer 1, we have now clarified this point in the legend of figure 3 which is the first forest plot in our manuscript, in page 6, (lines 183-186) “"Event" refers to the number of patients with a pCR for a determined HER2/CEP17 ratio. "Total" refers to the total number of patients for the same HER2/CEP17 ratio. In this case, for the same study, several HER2/CEP17 ratio thresholds were considered.”
Major comments:
Point 5: Several studies enrolled in this manuscript most probably reported different clinical-pathological characteristics. It would be interesting to meta-analyze some of these features to establish possible association with the categorization proposed for patients with HER2-overexpressing breast cancer (e.g. Stage, Grading, Tumor size). Then, the author can make a conclusion based on this result.
Response 5: Reviewer 1 raises an interesting point which is the possible link between HER2 amplification level, the clinical and biological characteristics of the tumor, and the pCR. In the manuscript we have submitted for review, we have reported some of these characteristics. We have now added a column for estrogen receptor status (ER). Unfortunately, much data is missing on the relationship between pCR and these different characteristics, and thus does not allow to perform such complementary analyses.
Point 6: Authors indicate that included patients with HER2-overexpressing breast cancer. Since they indicate overexpression is defined as ≥ 4 copies per nucleus and a HER2/CEP17 ratio ≥ 2, how were considered patient with ≥ 4 copies per nucleus but a HER2/CEP17 ratio < 2 or patient with < 4 copies per nucleus but a HER2/CEP17 ratio ≥ 2??. I think it is necessary to clarify this point (methods), overall when in the first paragraph of the page 8, it is explained why in most figures there is a class of categorization including cases without HER2-overexpression (ratio <2).
Response 6: Patients with ≥4 copies per nucleus and HER2/CEP17 ratio <2 correspond to patients with HER2 overexpression linked to polysomy 17. Only 1 study included such patients, the study by Greenwell et al. Patients with < 4 copies per nucleus but a HER2/CEP17 ratio ≥ 2 are rare, corresponding to patients with monosomy 17. In the studies included in our meta-analysis, no data related to such patients was available. That is why we had considered a Class-1 category with a ratio ≤ 2 (Figure 4). These 4,848 women included in the study by Greenwell et al. had HER2-overexpressing breast cancer with a HER2/CEP17 ratio ≤ 2 and HER2 copy number >6. They had a pCR of 29.2%, and when we excluded them, the new categorization in quartiles (Class-1:≥2, Class-2:≥3, Class-3:≥6 and Class-4:≥9) did not change the mean/median pCR in the different classes (Figure S2).
To follow the advice by Reviewer 1, we have now clarified this point in the methods section of our revised manuscript (lines 113-118) “Patients with ≥4 copies per nucleus and HER2/CEP17 ratio <2 correspond to patients with HER2 overexpression linked to polysomy 17. One study in the meta-analysis included such patients, who were classified in the HER2/CEP17 category ≤ 2. Patients with < 4 copies per nucleus but a HER2/CEP17 ratio ≥ 2 correspond to patients with monosomy 17, but no data related to such patients was available in the studies included in our meta-analysis”
Point 7: Regarding the above, do you consider that in these “equivocal cases” the treatment decision could be seen as influenced??. How do you correlate your results with this topic? How do your results may provide insights into the clarification of how these patients should be treated?? Some insight into the above should be included in the discussion section.
Response 7: Reviewer 1 raises an interesting point. Indeed, considering the conclusion of our meta-analysis, HER2 copy number should be considered instead of HER2/CEP17 ratio. However, most studies in our meta-analysis only provided the HER2/CEP17 ratio. Tumors with a polysomy 17 or a monosomy 17 have been considered in the consensus conference as the following sub-groups regarding HER2 IHC status and HER2 gene copy number. ISH group 2: (HER2/chromosome enumeration probe 17 [CEP17] ratio ≥ 2.0; average HER2 copy number < 4.0 signals per cell), ISH group 3 (HER2/CEP17 ratio < 2.0; average HER2 copy number ≥ 6.0 signals per cell), and ISH group 4 (HER2/CEP17 ratio < 2.0; average HER2 copy number ≥ 4.0 and < 6.0 signals per cell). Most of these tumors are usually considered as HER2 overexpressing tumors. [Wolff, Antonio C et al. J Clin Oncol. 2018]
In addition, low HER2 breast cancers (IHC scores of 1 and 2) are now considered for treatments using trastuzumab-deruxtecan at metastatic stages [S, Modi et al. N Engl J Med. 2022]. In our meta-analysis, we did not make any treatment recommendation. We suggest that for tumors with HER2/CEP17 ratio > 7, in the neoadjuvant setting, trastuzumab monotherapy could be considered as an alternative to dual therapy to decrease toxicities. In fact, tumors with a polysomy 17 usually do not have HER2 copy number >14 corresponding more or less HER2/CEP17 ratio >7.
Point 8: Authors declare that “If only the HER2 gene copy number was assessed, it was divided by 2 to obtain the HER2/CEP17 ratio”. For how many of the included studies do you did it?? What implications for analysis does this have? This must be considered a limitation that deserves to be highlighted in the discussion!!!
Response 8: Two of them only reported gene copy number, [Guiu et al. Br J Cancer. 2010 and Arnould et al. Clin Cancer Res. 2007] while two others only reported HER2/CEP17 ratio. That is why we chose HER2/CEP17 ratio for our meta-analysis, dividing by two the gene copy number of the two studies by Guiu et al. and Arnould et al. These articles included a total of 170 patients which only represents 1.5% of the total patient population (n=11238) included in the meta-analysis.
To follow the advice by Reviewer 1, we have now added this point in the discussion section of our revised manuscript (lines 236-242).
Point 9: “For the study by Greenwell et al., there were no details about the type of anti-HER2 therapy. We assumed that the women only received trastuzumab-based chemotherapy, since they were treated before 2013”. If this were not the case, do you think that the results provided by this study would be different? I suggest discuss this, together with the ideas of single vs dual blockade of HER2; just to better justify your point about toxicity
Response 9: Reviewer#1 raises an interesting discussion point.
pCR rates are significantly higher in case of dual anti-HER2 therapy compared to trastuzumab monotherapy [see the meta-analysis by Yu et al., J Clin Oncol 2020].
In our meta-analysis, out of a total patient population of 11238, 99 individuals (0.88% of the total) received dual anti-HER2 therapy combining trastuzumab and lapatinib, as part of their treatment. In the subgroup of patients with HER2/CEP17 ratio >7, this dual therapy was only given to some of the 13 patients of the study by Hurvitz et al. The detail is not provided in the original publication, but these 13 patients had a pCR of 85%. In a meta-analysis on 1,410 patients, dual HER2 blockade with trastuzumab and lapatinib led to pCR rates up to 63% [Guarneri V et al, ESMO Open 2022], suggesting that even in case of dual blockade, the HER2/CEP17 ratio may influence the pCR rate.
In conclusion, if most patients in our meta-analysis had received a dual anti-HER2 therapy, we think that the overall pCR rates would have been higher.
To follow the advice by Reviewer#1, we have now completed the discussion, page 9, lines 257 to 262: “In our study, in the sub-group of 1,089 patients with a HER2/CEP17 ratio above 7, up to 13 patients in the study by Hurvitz et al. were treated with a combination of trastuzumab and lapatinib, with a 85% pCR rate, suggesting that even in case of dual blockade, the HER2/CEP17 ratio may influence the pCR rate. In our meta-analysis, the pCR rate reached 79% when the HER2/CEP17 ratio was above 7, questioning the benefit of the dual blockade in this subgroup to avoid unnecessary toxicities”.
Reviewer 2 Report
Overall, the manuscript is well written to evaluate the relationship between HER2 amplification level and pCR rate in the neoadjuvant setting. In materials and methods, the reporting method and search algorithms are properly described. The authors suggest that the higher the level of HER2 amplification assessed by FISH, the higher the pCR rate would be in the neoadjuvant setting.
The authors point out that one of the limitations of this study is small sample size due to lack of relevant data reporting in majority of clinical trials. Particularly, the sample size of HER2/CEP17 ration > 7.0 is quite small and thus it seems to be difficult to conclude in this population. In addition, status of estrogen receptor expression is also a confounding factor which influences the pCR rate in neoadjuvant treatment for HER2-positive breast cancer.
Some typos and suggestions are pointed out in the separate file.
Below are my suggestions and questions.
1) Please consider including the proportion of ER-positive tumors in Table 1.
2) In figure 3, please indicate the differences between the same study names.
For example, there are seven “Wu Z et al. 2018” and two “Guiu S et al. 2010” counting events and totals. How did the authors divide the patient population as such? Is the way authors divided “Wu Z et al. 2018” into seven groups the same as they did “Guiu S et al. 2010” into two groups? Please indicate why they are reported as separate groups. Regarding this question, please consider adding the forest plot consisting of nine overall number of events (number of pCR) and number of total patients from nine studies to give readers the whole picture of the nine neoadjuvant studies if appropriate.
3) In figure 4, the authors reported that all the patients had pCR if HER2/CEP17 ratio is greater than 7.0. According to the original paper, it looks like there is one patient who had non-pCR despite HER2/CEP17 ratio exceeding 7.0. Please indicate why.
4) In figure 4, please indicate the differences of multiple but the same study names. (The same issue with 2))
5) In discussion, the authors state that pCR rate is very high when HER2/CEP17 is greater than 7.0 suggesting the possibility of overtreatment of dual HER2 therapy. Please indicate how many patients among the group with HER2/CEP17 exceeding 7.0 went through dual HER2 therapy.

Author Response
Response to Reviewer 2 Comments
Overall, the manuscript is well written to evaluate the relationship between HER2 amplification level and pCR rate in the neoadjuvant setting. In materials and methods, the reporting method and search algorithms are properly described. The authors suggest that the higher the level of HER2 amplification assessed by FISH, the higher the pCR rate would be in the neoadjuvant setting.
The authors point out that one of the limitations of this study is small sample size due to lack of relevant data reporting in majority of clinical trials. Particularly, the sample size of HER2/CEP17 ration > 7.0 is quite small and thus it seems to be difficult to conclude in this population. In addition, status of estrogen receptor expression is also a confounding factor which influences the pCR rate in neoadjuvant treatment for HER2-positive breast cancer.
Point 1: Some typos and suggestions are pointed out in the separate file.
Response 1: We have taken into account all these modifications in the revised version of our manuscript.
Below are my suggestions and questions.
Point 2: Please consider including the proportion of ER-positive tumors in Table 1.
Response 2: To follow the advice by Reviewer 2, we have included this information about the proportion of estrogen receptor (ER)-positive tumors in the articles that reported this data in Table 1.
Point 3: In figure 3, please indicate the differences between the same study names.
For example, there are seven “Wu Z et al. 2018” and two “Guiu S et al. 2010” counting events and totals. How did the authors divide the patient population as such? Is the way authors divided “Wu Z et al. 2018” into seven groups the same as they did “Guiu S et al. 2010” into two groups? Please indicate why they are reported as separate groups. Regarding this question, please consider adding the forest plot consisting of nine overall number of events (number of pCR) and number of total patients from nine studies to give readers the whole picture of the nine neoadjuvant studies if appropriate.
Response 3: To achieve the analysis of the data, we divided the patients into groups based on predetermined cutoff values specified in the relevant studies. The different event numbers with the same study names represent these different groups. To provide greater clarity, we have now included the specific cutoff values for each group in the Figure 3, in the revised version of our manuscript.
Point 4: In figure 4, the authors reported that all the patients had pCR if HER2/CEP17 ratio is greater than 7.0. According to the original paper, it looks like there is one patient who had non-pCR despite HER2/CEP17 ratio exceeding 7.0. Please indicate why.
Response 4: Reviewer 2 is right. However, in the study by Wu et al. the published graph presents an interval of 2 by 2 (2, 4, 6, 8, …) suggesting that the sensitivity analysis of cutoff goes from 2 to 2. In our meta-analysis we chose the cutoffs on the basis of sensitivity analyses: we categorized the studies in quartiles, as follows: ≤ 2 (class 1); 2.1 to 5.0 (class 2); 5.1 to 7.0 (class 3); and > 7.0 (class 4). In the study by Wu et al., there is no threshold at 7, so that the patient with pCR close to 8 but less than 8 was included in the subgroup with the cutoff ≥6. Overall it does not change the results of our meta-analysis.
Point 5: In figure 4, please indicate the differences of multiple but the same study names. (The same issue with 2))
Response 5: To provide greater clarity, we have also included the specific cutoff values for each group in the revised version of the Figure 4.
Point 6: In discussion, the authors state that pCR rate is very high when HER2/CEP17 is greater than 7.0 suggesting the possibility of overtreatment of dual HER2 therapy. Please indicate how many patients among the group with HER2/CEP17 exceeding 7.0 went through dual HER2 therapy.
Response 6: Reviewer 2 raises an interesting point. Out of a total patient population of 11238, 99 individuals (0.88% of the total) received dual therapy as part of their treatment. In the subgroups of patients with HER2/CEP17 ratio >7, a dual anti-HER2 therapy combining trastuzumab and lapatinib was only given to some of the 13 patients of the study by Hurvitz et al. The detail is not provided in the original publication, but these 13 patients represent 1.2% of the patients in this subgroup, and had a pCR of 85%. In a meta-analysis on 1,410 patients, dual HER2 blockade with trastuzumab and lapatinib led to pCR rates up to 63% [39], suggesting that even in case of dual blockage, the HER2/CEP17 ratio may influence the pCR rate.
Round 2
Reviewer 1 Report
Authors have considered most of the review comments and included clear insights regarding discussion points
Author Response
Comment 1 : Authors have considered most of the review comments and included clear insights regarding discussion points
Answer 1 : Thank you for your valuable feedback and for taking the time to review our work. We are grateful to hear that you find our revisions clear and that we have addressed most of your comments.